# Establishing Age-Based Color Changes for the American Burying Beetle, *Nicrophorus americanus* Olivier, with Implications for Conservation Efforts

**DOI:** 10.3390/insects14110844

**Published:** 2023-10-31

**Authors:** Robert Shane McMurry, Michael C. Cavallaro, Andrine Shufran, William Wyatt Hoback

**Affiliations:** Department of Entomology and Plant Pathology, Oklahoma State University, Stillwater, OK 74078, USA; shane.mcmurry@okstate.edu (R.S.M.); michael.cavallaro@okstate.edu (M.C.C.); andrine@okstate.edu (A.S.)

**Keywords:** Silphidae, cuticular color, age determination, insect aging

## Abstract

**Simple Summary:**

The American burying beetle, *Nicrophorus americanus*, is federally protected as a threatened species because of population decline in most of its historic range in North America. Recovery of the species requires observation and cataloging of individuals captured in the wild. Previous work has characterized adult beetles as teneral (young) or senescent (old) but improving age determination of adults that survive approximately 90 days is critical to future recovery outcomes. As an adult, the beetle has distinctive orange/red markings over a black background. Because a demonstrable color difference was observed under laboratory conditions, we determined that a gradual, age-based color gradient exists and can be visually assessed to estimate the age of an individual beetle within 15-day intervals. We photographed adult *N. americanus* at set time intervals over their lifespans and generated a pixel-based approach to age adult beetles in a nondestructive manner. The orange/red color near the head became predictably darker as the beetles aged. A color chart was created that should allow conservation practitioners to assign beetles to six distinct age categories in the field without requiring additional equipment. This technique will refine estimates of population structure and improve the selection of individuals for breeding, potentially improving reintroduction program success.

**Abstract:**

The American burying beetle, *Nicrophorus americanus* Olivier, is a federally protected insect that once occupied most of eastern North America. Adult beetles feature distinct, recognizable markings on the pronotum and elytra, and color changes with age have been observed. Among the challenges faced by research scientists and conservation practitioners is the ability to determine beetle age in the field between and including teneral (young) and senescent (old) adult stages. Using 20 (10 male and 10 female) captive-bred beetles, we characterized the change in greyscale and red, green, and blue (RGB) color channels over the lifespan of each beetle for field-aging applications. Individual beetles were photographed at set intervals from eclosion to death, and color data were extracted using open-source ImageJ Version 1.54f software. A series of linear mixed-effects models determined that red color showed the steepest decrease among all color channels in the pronotum and elytral markings, with a more significant decrease in the pronotum. The change in greyscale between the pronotum and elytral markings was visibly different, with more rapid darkening in the pronotum. The resulting pronotum color chart was tested under field conditions in Oklahoma, aging 299 adult *N. americanus*, and six age categories (day range) were discernable by eye: teneral (0–15), late teneral (15–31), early mature (31–45), mature (45–59), early senescent (59–76), and senescent (76–90). The ability to more precisely estimate age will improve population structure estimates, laboratory breeding programs, and potential reintroduction efforts.

## 1. Introduction

The American burying beetle, *Nicrophorus americanus* Olivier, previously ranged across eastern North America and into parts of Canada [1,2]. However, numbers and distribution have declined—by as much as 90%—with remaining natural populations currently recorded in only six states: Rhode Island, South Dakota, Nebraska, Kansas, Oklahoma, and Arkansas [1,3,4]. Because of its decline, *N. americanus* was listed as federally endangered in 1989 when very isolated small populations were known [5]. It was reclassified as threatened in 2020 by the Fish and Wildlife Service based on information that suggested a diminished threat of extinction in its current range [6]. As part of recovery efforts, attempts to restore *N. americanus* to areas where it once occurred have been underway since 1991, with varying success [7,8]. Monitoring and mark–recapture surveys continue to play an important role in *N. americanus* conservation efforts, with extant populations primarily threatened by climate change, agricultural intensification, changes to carrion resources, and interspecific competition [4,8].

Like many *Nicrophorus* spp., *N. americanus* has distinctive orange/red markings over a black background. These red colors occur in similar places on the body for most *Nicrophorus* spp.: the antennal club; less ubiquitously, the clypeus; and in mostly separate patches on the anterior and posterior of the elytra, as well as the distal margins of the elytra [2]. Unique among North American species, *N. americanus* also has a large mark on its pronotum [9]. Age grading was utilized by Bedick et al. [5] by noting the clypeal membrane hue and brightness to document *N. americanus* life history in Nebraska. Three age categories were designated: mature for beetles that had emerged in June after overwintering, teneral for beetles that were newly produced within the year, and senescent for beetles that were at the end of their lifecycle in the current year. The *N. americanus* is thought to be univoltine across its range, where new adults emerge in fall and overwinter as adults, then reproduce the following year before senescing and dying [5,10].

Functionally, contrasting colors on burying beetles have been shown to provide several benefits, including aposematic warning [11] and sexual signaling of immune function [12]. The clypeal membrane hue and brightness have been associated with immune response in *N. pustulatus* and were found to be positively correlated with age; the more red (lower hue values) and darker (lower brightness values) membranes indicated an increased immune response in both sexes [12]. While qualitatively noted by some, there has yet to be a study examining *N. americanus* pronotum color and its association with age, and, moreover, little published data exist on how long *Nicrophorus* beetles, specifically *N. americanus*, survive in captivity and in the wild. Better resolution of the age of field-collected beetles can provide a predictive measure of brood production and an assessment of life history across their range, overwinter survival, and potential to reproduce [5].

We have maintained a colony of *N. americanus* at Oklahoma State University (Permit no. TE-045150-4.1) and have noted the progressive change in color of the orange/red patches over time. Immediately after eclosion from the pupa, the patches appear bright orange, and then incrementally shift toward a darker red until death. After death, individuals quickly turn to a dark brown with decomposition. This occurs within both wild-caught and lab-reared individuals, irrespective of the diets upon which they are maintained. In the laboratory, the color changes occur in both individuals that are used for breeding that burrow into a substrate, as well as those that are never given access to a soil-based substrate. Collectively, observations made in the field and laboratory lead to the objectives of the current study.

Here, we characterized the rate of color and greyscale change in individual, captive-bred *N. americanus* over time with a known date of emergence. Our objectives were to determine the viability of nondestructive age grading for wild-caught *N. americanus*, explain potential variability from beetle sex, weight at emergence, and geographic origin, and compare the difference in visible color change between distinct pronotum and upper elytral maculations. From these pixel-based color data, we generated and tested a color chart during summer field sampling of *N. americanus* in Oklahoma and present the results. We expected to create an improved field age-grading system based on characteristics visible with the naked eye for wild-caught *N. americanus*.

## 2. Materials and Methods

### 2.1. Rearing

Rearing procedures were adapted from unpublished care instructions received from the Center for American Burying Beetle Conservation at the St. Louis Zoo in St. Louis, Missouri. All *N. americanus* recorded in these experiments were the first generation of wild-caught parents and were reared in a basement at Oklahoma State University’s Insect Adventure, where the colony is kept at 24.4–25.5 °C. Brood buckets were constructed using 19 L buckets with 4 mm holes drilled into the bottom to allow water drainage and 140 mm holes cut into the lid for air flow. Both the inside bottom and inside lid hole were covered using a fine metal screen attached with hot glue to prevent escape. The outside of the lid hole was covered in synthetic fine-mesh organdy cloth attached with hot glue to prevent nuisance fly access.

Brood chamber substrate was composed of 3 parts Greensmix^®^ sphagnum peat moss to 1 part sandy loam topsoil obtained from the banks of the Cimarron River near Perkins, Oklahoma. The substrate was completely saturated using hot tap water and allowed to drain and settle for 7 days. Once the substrate had sufficiently drained, a previously frozen quail (purchased from Rodentpro Premium Animal Nutrition, Inglefield, IN weighing approximately 110 g was thawed and then placed on top and in the center of the substrate. One male and one female *N. americanus* were introduced onto the substrate and the lid was closed. Buckets were checked once a day for 3 days. If the quail carcass was completely or partially buried, both beetles were left in the bucket and allowed to continue their reproductive behaviors for 10 days, at which time the male was removed and the female was left with the larvae. At 15 days post-introduction, the female was removed and a visual search for larvae was conducted. If larvae were found, they remained undisturbed for 50 days and then were checked daily until new adult beetles surfaced. New beetles were sexed, weighed, and measured for pronotal width, before they were separated into individual containers. Throughout the experiment, the beetles were kept individually in semi-translucent plastic containers (76 mm × 78 mm depth) with a damp paper towel. They were fed three times a week (i.e., Monday, Wednesday, and Friday) with two mealworms, *Tenebrio molitor* L., and two greater waxworm larvae, *Galleria mellonella* L.

### 2.2. Digital Imaging

For practical field age assessments of a federally threatened insect, a pixel-based approach to color data extraction was chosen over reflectance spectra measurements via spectrophotometry. Previous studies confirmed that the use of a spectrophotometer to measure live insects ex situ is not practical or feasible in the field [13]. For the purpose of field-based applications, and to encourage accessibility for researchers, conservation practitioners, and environmental consultants, we used methods modified from Hartmann et al. [14] that paired digital photography with open-source Fiji/ImageJ Version1.54f computer software [15].

Initially, digital photographs were taken of four *N. americanus*, two males and two females, from the laboratory colony originating with Oklahoma-collected beetles. Methods were refined and standardized and an additional 16 *N. americanus* (8 males and 8 females: half from Oklahoma parents and half from Nebraska parents) from the lab colony were photographed—from emergence until death. Pictures were taken with a Moto G Power 2022 android smartphone (specifications: 50 MP sensor, 1.3 µm (Quad Pixel) f/1.8 2 MP macro, 1.75 µm f/2.4 2 MP depth 1.75 µm f/2.4, Single LED flash). Beetles were photographed three times a week under ambient LED lighting on a standardized staging area, which was an off-white laminated work bench. A camera stand made from PVC was used for consistency, beginning at day 40 during the initial trial with four beetles, and was used for all subsequent photographs. The stand was made of 20 mm PVC that was 21.5 cm long, using four three-way right-angle connectors as the corners, with 11.4 mm end caps as the base. The frame measured 130 mm tall, 210 mm long, and 117 mm wide. The inner dimensions used to secure the phone were 155 mm long by 64 mm wide. The camera was 12.5 cm from the beetle, creating a field of view of 14.8 cm × 11.1 cm.

All beetles were secured between two fingers and photographed in the dorsal view, with the pronotum, elytra, and head visible in the same frame. The photographs included in-frame, the individual ID, sex, and date recorded on a laminated card, so that the number of days since emergence could be determined. After the pictures were taken, the beetles were placed back into their containers until the next photograph date.

The red, green, and blue (RGB) values were obtained for each beetle using open-source Fiji/ImageJ computer software developed by the U.S. National Institutes of Health [15]. Each full image was uploaded and the rectangle tool was used to draw a 25 × 25 pixel square (1 mm × 1 mm) that was positioned at consistent locations on the beetle body. For the pronotum, the nonblack portion adjacent to the center of the head was selected. This location was chosen to avoid glare from the camera flash. For the elytral RGB values, the 25 × 25 pixel square was aligned to the top-left portion of the upper elytron when viewed dorsally. ImageJ generated a color histogram for each 25 × 25 pixel square sampled. RGB values were collected for each location and recorded in a spreadsheet. Extracted RGB values were used to calculate greyscale. The laminated background had mean RGB color values (±SE) of 146 ± 2, 139 ± 2, and 144 ± 3, respectively. This was performed for every photograph taken over the lifespan of each individual beetle.

### 2.3. Field Tests

We tested the developed color scale as part of ongoing monitoring of the population of *N. americanus* at Camp Gruber, a military training facility, located near Braggs, Oklahoma. The facility is an Army National Guard training facility occasionally used by other military personnel and police task forces and is open to seasonal public hunting. Located in Muskogee County, OK, Camp Gruber encompasses approximately 22,500 ha (55,680 acres). The base has a cantonment area (250 ha) and firing ranges (185 ha), with the remainder being a mixture of grassland and forest with streams and ponds, with the southeastern corner bordered by Greenleaf Lake (370 ha). Sampling for *N. americanus* occurs with above-ground pitfall traps baited with previously frozen whole laboratory rats. Twenty traps were placed with a minimum of 2.2 km spacing and monitored for a 5-day period, with bait replacement on day 3. Sampling occurred on 11–15 July, with age-rating beginning on 12 July and 15–19 August. All captured *N. americanus* were assessed for age by comparing the pronotum with a printout of the developed digital scale, using the six new age categories (day range): teneral (0–15), late teneral (15–31), early mature (31–45), mature (45–59), early senescent (59–76), and senescent (76–90). Age was assessed by two individuals for each beetle and recorded after agreement.

### 2.4. Data Analyses

Statistical analyses were conducted using R version 4.1.1 [16]. A series of linear mixed-effects models (LMMs package lme4 [17]) were used to assess the change in the red, green, and blue color channels and greyscale over time from distinct markings on the pronotum and elytra of 10 male and 10 female *N. americanus*. The model set used to determine the change in color channels and greyscale over time included the fixed effect: beetle age (number of days), and the model set used to characterize the comparative change in distinct markings from the pronotum and elytra included the fixed effects: beetle age + beetle marking. The comparative model set was also evaluated as an interaction term: beetle age * beetle marking. After these analyses, we used the color channel and distinct marking that produced the greatest change over time as the response variable with the fixed effects: sex (m/f), beetle weight at emergence (g), and origin (Oklahoma or Nebraska). Random effects included photography technique and beetle identity and were used to account for potential variation in each model set. Three different photography techniques were used over the entire age-based color assessment. During the refinement stage, two techniques were used with the first four beetles, where two different people held the camera approximately 12.5 cm from the insect. The third technique used the camera stand as described and all photographs were taken by the same person. Using a chi-square distribution, a direct comparison with likelihood ratio tests (LRT) among models determined the significant variation over time (*p* ≤ 0.05).

## 3. Results

On average (±SD), the 20 beetles lived 66 ± 24 days (range 11 to 99), with 5 beetles living more than 90 days (SD = 4). Measurements taken from the pronotum and elytron of each beetle on the same 3 days every week revealed a significant change in color during the lifespan of the photographed beetle population (Figure 1). Each measured color channel decreased in value. Red color significantly decreased over time in the pronotum (β ± SE = −12.1 ± 0.6, *p* < 0.001) and elytral markings (β ± SE = −9.8 ± 0.7, *p* < 0.001), displaying the greatest decrease among all color channels (Table 1). Red color models had the greatest adj. *R*^2^ values among all color channels, which further supports the best fit in the data series over time. The random effect (photography technique) generated the greatest variance in red color among the pronotum and elytra models, with a variance ± standard deviation of 190 ± 14 and 125 ± 11, respectively (Table 1). The adj. *R*^2^ for the pronotum and elytra red color null models were 0.69 and 0.58, respectively.

Green color significantly decreased over time in the pronotum (β ± SE = −6.6 ± 0.4, *p* < 0.001) and elytral marking (β ± SE = −6.1 ± 0.5, *p* < 0.001). However, the change in green color was less steep than red color and it was not significantly different over time ([LRT]; χ2(1) = 0.0, *p* > 0.05), indicating the potential importance of decreases in both color channels. Blue color significantly decreased over time, albeit to a lesser degree in the pronotum (β ± SE = −1.2 ± 0.3, *p* < 0.001) and elytral markings (β ± SE = −1.7 ± 0.4, *p* < 0.001). Blue channels were the least influenced by the random effect. Greyscale is calculated from RGB color values and, accordingly, displayed similar significant decreases in the pronotum (β ± SE = −6.9 ± 0.4, *p* < 0.001) and elytral markings (β ± SE = −5.7 ± 0.5, *p* < 0.001), becoming darker over time (Figure 2).

Comparative change in the color channels from the pronotum and elytral markings were significantly different for red color (Table 2). The interaction term, i.e., beetle age * beetle marking, performed better than the additive model and was retained ([LRT]; χ2(1) = 9.5, *p* = 0.002). Red color had the greatest measured difference between beetle markings, decreasing approximately 19% faster in the pronotum (β ± SE = −1.9 ± 0.6, *p* = 0.002). Decreases in the green and blue color values over time were less in the pronotum than the elytra. Greyscale values were not significantly different between the pronotum and elytral markings; however, the pronotum became darker 17% faster than the elytral markings (β ± SE = −0.75 ± 0.5, *p* = 0.10).

Similarly, to the marking-specific models (i.e., pronotum and elytra), the random effect generated the greatest variance in the red color values. Using the red color values collected from the pronotum markings, the variance in beetle sex, weight at emergence, and origin were evaluated. The adj. *R*^2^ for beetle sex, weight at emergence, and origin were 0.01, <0.01, and 0.08, respectively. Origin (Oklahoma or Nebraska) explained the greatest variation among these factors but was not significant (χ2(1) = 0.09, *p* = 0.77). Variance explained by beetle sex and weight at emergence were also not significant (*p* > 0.05).

For field validation, a total of 366 *N*. *americanus* were captured over the 9-day period of assessment (Table 3). In July, the majority of beetles were newly emerged, suggesting that reproduction had occurred approximately two months prior. As predicted by the laboratory results, in August, the majority of captured beetles were older than in July. From July to August, the percentage of teneral and late teneral beetles decreased by 33%, and the percentage of early mature and mature beetles increased by 26%.

## 4. Discussion

In this study, we measured the change in RGB color channels and greyscale of markings among 20 laboratory-reared *N. americanus* over their entire adult lifespans. During this period, we quantified the predictive darkening of orange-colored pronotal and elytral markings. RGB color values decreased in value significantly over time, which translated to overall changes in greyscale. The rate of darkening differed between markings, with the pronotum becoming significantly darker at a faster rate than the elytral marks, suggesting that it can most reliably be used to estimate beetle age (Figure 1 and Figure 2). We validated the use of the generated color chart under field conditions by aging wild-caught beetles. Together, the predictive darkening of adult markings paired with beetle age highlight the use of age-based color change to improve conservation.

### 4.1. Biotic and Abiotic Influences on Color Change

As *N. americanus* aged, their pronotum color transformed from bright orange to a darker red (Figure 1). Insect color visible to humans and instrumentation are broadly classified as either pigmentary or structural [18,19]. Pigmentary color is used in many forms of signaling between organisms of the same species (sexual or dominance) and between different species, most often as warning coloration [20,21]. Pigments can be grouped based on either derivation or precursors, i.e., those that are synthesized by the insect de novo or those acquired from the diet. Within pigments, the carotenoids are often the most common red and orange pigment category; however, in animals, these are almost always diet-derived [22].

The source of color on *N. americanus* pronotum and elytra is unknown and, to date, no study has determined if the color in any *Nicrophorus* spp. is pigment-based. Some studies have measured hues from *N. americanus* and related *Nicrophorus* species, e.g., [12]; however, none have identified the source of color on a physical or molecular level. Anecdotally, black paired with red or orange, like that displayed by *N. americanus*, creates high contrast, and suggests that the markings are meant to be highly visible. The gradual and irreversible darkening with age indicates that the colors are most likely pigmentary rather than structural. Although it is possible that microstructures on the cuticle could change color for a burrowing species through surface abrasion, the color shift also occurs in individuals that never have access to soil. Further, the rapid color change (blackening) at death likely indicates a catabolic change to pigmentary molecular structure, although desiccation has been found to reversibly change structural color in tortoise beetles [23,24]. The red coloration in elytral marks can be altered with a cauterizer for marking [25], suggesting that either the pigments are heat sensitive or the pigmentary color is influenced by surface structure.

Within pigmentary candidates for *N. americanus* orange color, some seem more likely than others. Although carotenoids are often used by diverse species in color signaling, there is a significant obstacle against their use by carrion beetles. Burying beetles, including *N. americanus*, are necrophores and insect predators, and are therefore not likely to frequently acquire carotenoids through their diet. Thus, the pigments responsible for the orange-to-red patches on *N. americanus* are unlikely to be carotenoids unless the pigments are formed through symbiotic microbial synthesis, as observed in some plant-feeding hemipterans [26,27]. Being similarly plant-diet restricted, flavonoids are also unlikely to be responsible for the coloration [21]. Pterins are synthesized without the requirement for plant diets, but they have yet to be identified as a pigment in beetle elytra [28]. More likely candidates include ommochromes, which have been shown to change from a more yellow to red form across an individual dragonfly’s lifespan [29], and melanins.

Melanin in the form of pheomelanin seems the most likely candidate for *N. americanus* orange–red pigmentation. Wormington and Luttbeg [12] evaluated the correlation between clypeal color and immune function of *Nicrophorus pustulatus* by injecting beetles with sandpaper in their abdominal cavity to trigger an immune response, then analyzing the color change of the clypeus. They found that manually triggering an immune response in *N. pustulatus* resulted in a significantly darker clypeus color, leading the authors to suggest that the measured pigment was pheomelanin. The darkening by age could be a result of immune challenges, as beetles have a microbiome that is used for carcass preservation [30]. Determining the identity of the pigment in question may help to illuminate some of the selection pressures faced by *N. americanus* and other burying beetles, along with further elucidating their potential use as a signal mechanism. Because *N. americanus* are exposed to many microbes through carcass interaction, darkening could also indicate immune competency [12].

Data generated in the current study were obtained from beetles bred and reared under laboratory conditions. Under these controlled environmental conditions, a base inherent rate of change was established without the potential variation induced by differing environmental and behavioral conditions. Future studies will hopefully expand on the varying abiotic factors that influence color change. In our study, beetles were reared under a stable 12L/12D photoperiod and nearly constant temperature of 25.5 °C (78 °F). Bedick et al. [5] found that *N. americanus* in Nebraska were active at temperatures between 12.7 °C (54.86 °F) and 24 °C (75.2 °F), while adults in diapause presumably experience winter temperatures very near freezing [31]. The upper limit may be a consequence of *N. americanus’s* susceptibility to desiccation, as found for *Nicrophorus marginatus* [32]. If *N. americanus* are particularly susceptible to desiccation, influenced by the structure of their cuticle, then variation in aridity could play a part in the appearance of their colored spots. Under controlled conditions, beetles were only subjected to direct fluorescent lighting during the brief period when they were photographed and fed, approximately 3 min total weekly. Under natural conditions, these largely nocturnal beetles would rarely be exposed to sunlight, but even brief exposure to ultraviolet light could affect color through the breaking of chemical bonds.

*N. americanus* are beneficial to the environment by reducing fly breeding resources, decomposing organic matter, cycling nutrients, and increasing soil fertility [33] through burying carrion and rearing their young together—a rare trait among invertebrates [34]. Complex social interactions in invertebrates can contribute to greater longevity [35], as can the reliance on ephemeral resources that are patchy and unpredictable. In our study, we did not allow for intraspecific competition, mate-pairing, or reproduction, and these behaviors may result in abrasions that change the beetle cuticle [5]. *N. americanus* males typically fight other intraspecific males at a carcass site [2] and must contend with interspecific competition from other attracted invertebrates [36] while escaping predation risk from vertebrates [37].

Both sexes that engage in reproduction have to bury the carcass and form their brood ball by pushing through the soil substrate, which is frequently hard and full of dense grass roots in areas where many *Nicrophorus* spp. must partition habitat [34]. *N. americanus* can utilize variable soil textures for brood rearing but appear to show preference for high proportions of sand [1], possibly lowering bulk density for faster burial [34] and/or increasing oxygen diffusion. Moving through sand could, in particular, accumulate many small or large abrasions, causing changes to the perceived color on their external markings. Additionally, *N. americanus* is host to phoretic mites on their cuticle that can behave in a mutualistic manner by reducing fly eggs on the carcass [38]. These mites may also cause abrasive wear from their active movements and high densities.

Observations made in the field study by Bedick et al. [5] show that senescent beetles also had greater changes in coloration of the pronotum compared to the elytra, and older beetles showed obvious signs of damage. Similarly, Jenkins et al. [25] distinguished teneral and senescent beetles while developing a marking technique that uses a cauterizer to ablate the elytral mark. This marking technique is permanent and removes the orange color in the area exposed to the cauterizer, suggesting that surface structure plays a role in coloration. Darkening with age in the absence of abrasions or other damage could aid in mate choice. A darker pronotum might signify an older female beetle, potentially making her more attractive to a mate as age correlates with offspring provisioning in *Nicrophorus orbicollis* [39]. Older females allocate less of the carrion to their own body mass, reserving the bulk of the resource for their offspring and shortening their lifespan. Further, females were found to produce larger broods and larger eggs when they were 65 days old [39].

### 4.2. Age Grading, Colorimetry, and N. americanus Color Chart

Accurate age grading is valuable to the understanding of insect ecology and behavior, allowing for more detailed assessment of the effects of climate change on the age structure of an insect population. Age-grading techniques also allow researchers to predict generations per year and provide insights into broods per individual, refining estimates of population viability and intrinsic rate of growth. In addition, records of survival in captivity and the aging process provide insights into captive breeding programs and may improve reintroduction efforts. Together, the improved knowledge of a population age structure makes it possible to more accurately predict population changes over time [40].

However, age grading of insects is often a complicated task as age-related fitness traits are sensitive to a range of environmental factors, including temperature, humidity, diet, and oxygen levels [40]. Research on aging typically relies on experimental cohorts of individuals of known age, or on physiological, morphological, or behavioral traits assumed to vary with age [41]. As a result, lab-produced specimens can differ greatly from those caught in the wild, adding another complication to insect-aging research. Multiple studies have been conducted to determine the best indicator for insect aging with varying results, e.g., [35,42].

Colorimetry, the study of the physical and perceptual aspects of color, has a long history outside of biological applications [43]. More recently, biologists interested in field applications have attempted to use technological advancements to overcome the problems associated with standardizing the human perception of color. Acknowledging the factors associated with using color in biological fieldwork, below, we outline the applicability of our data to age grading *N. americanus* to improve monitoring and conservation. We also created a *N. americanus* pronotum time-series color chart (Figure 3) that may allow further field testing of populations under different conditions.

For the purposes of applying our results to the age grading of wild-caught individuals, a color guide needs to be useful in the following ways: First, the chart must show a representation of color perceivable and accurate to the human eye, with enough difference between age-class charts that users with varying color sensitivity can readily perceive the differences. Second, it must be useful under varying light conditions, or else adjusted for the most common lighting for field sampling sites and season. If it is necessary to limit its use to a narrowly defined set of outdoor conditions, they must be clearly communicated for end users (e.g., in full sun between certain hours post-sunrise). Finally, if it is printed, a way to test the representative color charts under controlled conditions must be described so that charts where print pigments have changed significantly can be replaced with a newly printed copy. This can be performed by eye in the lab against a computer screen, as described by Aguiar [44].

In Oklahoma, conservation practitioners and environmental consultants take digital images of captured *N. americanus* for the purposes of size measurement. Adding an inexpensive color standard chart/calibration standard in-frame to correct for different lighting conditions could be used so that field age estimates could be further refined with later analysis [45,46,47,48]. It is important to note that digital image files must be saved and analyzed in RAW format, to avoid compression or other transformation-related color changes [48].

Field testing was accomplished with printouts and digital images on cell phones. Beetles could be assessed for age under variable field conditions (sampling takes place between sunrise and 10:00 A.M.), and the color chart improved the conventional designations of teneral (young) and senescent (old) beetles into six distinct age categories. The data showed a progression in July of the majority of individuals being newly emerged to the majority of beetles maturing in August (Table 3). These data provide better demographic information that can be used to assess reproductive period and adult biology and greatly improve the understanding of population characteristics beyond aging methods developed by Bedick et al. [5].

### 4.3. Future Directions

Categorizing the relative age of wild-caught *N. americanus* with a color gradient chart provides researchers and conservation practitioners with a multifaceted age-grading tool. Continued conservation efforts for *N. americanus* will rely on captive breeding paired with reintroduction programs, where color can play a critical role in captive mate pairing and the subsequent release of beetles at the most appropriate age to produce successful broods. Bedick et al. [5] suggested that senescent adults, identified by their darkened pronotal markings, might have been past reproductive capability and unsuitable for establishing captive breeding colonies. Beyond the ability to discern reproductive adults, color age grading allows for the assignment of wild populations into more precise age groups and generates data for demographic studies. Life table analyses can determine adult beetle age, or an age range, that experiences high mortality rates, and risk assessments can focus on identifying mortality factors with a clear application to conservation efforts [35].

Our study produced a baseline color gradient that links a measured RGB value on the pronotum and elytra by age. Pronotum or elytra color gradient charts should be validated for field use by comparing the measured colors from captive and wild beetles. Further optimization could include age-grading mark–recapture studies or measures of captive-bred beetles in ex situ cage field assessments, investigating the influence of environmental variables on color change. As technology improves, the age-grading technique may be able to be improved using relatively inexpensive field-portable spectrophotometers [13]. However, any use of a spectrophotometer would need to be optimized to take a reading on a live insect that can quickly be released unharmed. Ultimately, we believe that a print or electronic color chart will allow researchers and conservationists to adopt better age grading and easily estimate adult American beetle age in a nondestructive manner.

## Figures and Tables

**Figure 1 insects-14-00844-f001:**
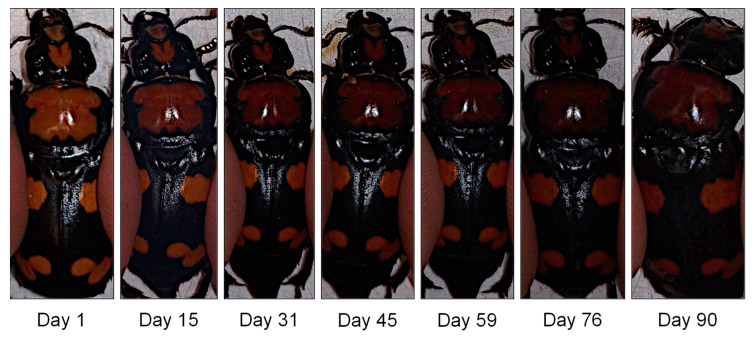
Representative example of changes in adult *Nicrophorus americanus* (male) cuticular coloration between the two distinct markings on the pronotum and elytra over time (days since emerged).

**Figure 2 insects-14-00844-f002:**
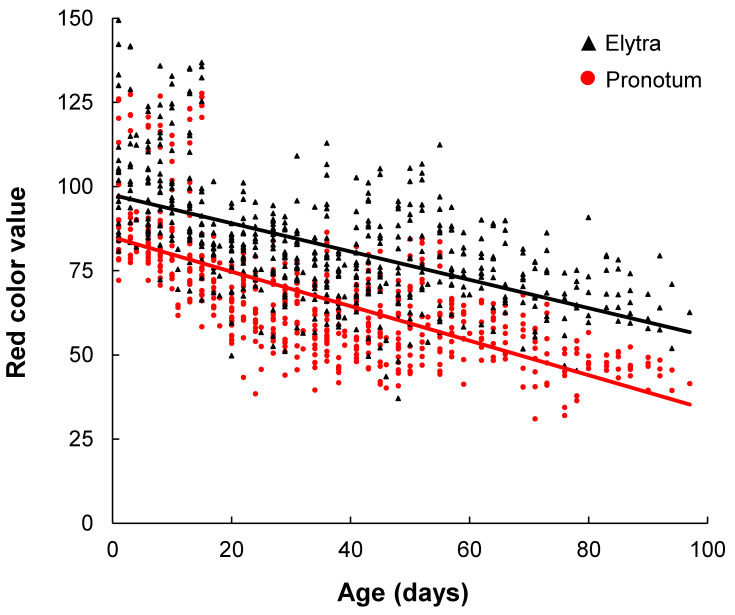
Measured change in red color from the pronotum and elytral markings over time among the 20 adult *Nicrophorus americanus*. Pronotum red color decreased at a faster rate than the elytral markings (β ± SE = −1.9 ± 0.6, *p* = 0.002). Red color values follow the color intensity scale, where 0 is the lowest intensity and 255 is the highest.

**Figure 3 insects-14-00844-f003:**
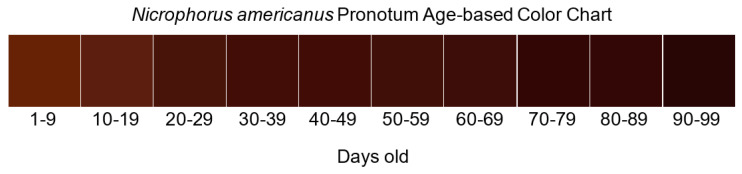
Time-series gradient of mean RGB values measured from adult *Nicrophorus americanus* over time.

**Table 1 insects-14-00844-t001:** Results of LMMs assessing the change in greyscale and the red, green, and blue color channels (response variable) from color markings on the pronotum and upper elytron maculation over the lifespan of each beetle (fixed effect).

Beetle Marking	Model Parameters ^a^								
								σ^2^_i_	
Pronotum	AIC	∆AIC	*R* ^2^	β	SE	χ2	*p*	PT	BID
Greyscale	3592.2	238.9	0.59	−6.9	0.4	322.6	<0.001	56.5 ± 7.5	14.5 ± 3.8
Red	4087.1	280.9	0.69	−12.1	0.6	390.6	<0.001	295 ± 17	52.8 ± 7.3
Green	3613.4	223.0	0.57	−6.6	0.4	289.8	<0.001	12.2 ± 3.5	10.2 ± 3.2
Blue	3438.9	18.5	0.04	−1.2	0.3	22.4	<0.001	0.32 ± 0.6	0.0 ± 0.0
Elytra	AIC	∆AIC	*R* ^2^	β	SE	χ2	*p*	PT	BID
Greyscale	3962.9	99.2	0.38	−5.7	0.5	113.5	<0.001	14.4 ± 3.8	17.3 ± 4.2
Red	4274.5	150.0	0.58	−9.8	0.7	183.6	<0.001	132 ± 12	75.5 ± 8.7
Green	4022.9	100.8	0.36	−6.1	0.5	133.8	<0.001	2.3 ± 1.5	14.7 ± 3.8
Blue	3924.1	11.1	0.02	−1.7	0.4	15.7	<0.001	3.9 ± 1.9	0.0 ± 0.0

^a^ Model parameters are listed by column: Akaike’s Information Criterion (AIC), change in AIC (∆AIC) from the intercept-only model, adjusted coefficient of determination (*R*^2^), beta coefficients (β), standard error of β, chi-square (χ2), probability value (*p*), and variance of random effect (σ^2^_i_). Covariate in model structure: beetle age (number of days). Random effects: photography technique (PT) and beetle identity (BID).

**Table 2 insects-14-00844-t002:** Results of LMMs assessing the change in greyscale and the red, green, and blue color channels (response variable) explicitly testing the interaction of color markings on the pronotum and upper elytron maculation over time (i.e., beetle age * location of distinct beetle marking).

Response Variable	Model Results ^a^								
								σ^2^_i_	
	AIC	∆AIC	*R* ^2^	β ^b^	SE	χ2	*p*	PT	BID
Greyscale	7608.8	535.5	0.55	−0.75	0.5	2.67	0.10	35.5 ± 5.9	14.6 ± 3.8
Red	8379.5	765.2	0.69	−1.99	0.6	9.54	0.002	214 ± 15	60.8 ± 7.8
Green	7733.9	523.3	0.53	−0.01	0.02	0.30	0.58	3.6 ± 1.9	9.5 ± 3.1
Blue	7472.6	23.9	0.03	−0.00	0.02	0.002	0.96	3.8 ± 1.9	0.42 ± 0.6

^a^ Model parameters are listed by column: Akaike’s Information Criterion (AIC), change in AIC (∆AIC) from the intercept-only model, coefficient of determination (*R*^2^), beta coefficients (β), standard error of β, chi-square (χ2), probability value (*p*), and variance of random effect (σ^2^_i_). Covariate in model structure: beetle age (number of days). Random effects: photography technique (PT) and beetle identity (BID). ^b^ Beta coefficients reflect the comparative estimates in the pronotum (i.e., the greyscale and color channels measured on the pronotum decreased more rapidly than on the elytra).

**Table 3 insects-14-00844-t003:** Color-based ages of 366 field-collected *Nicrophorus americanus* (189 male and 177 female) collected from eastern Oklahoma in 2023.

Dates Sampled	Day Range	New Age Category	Number of Adults
12–15 July	0–15	Teneral	73
	15–31	Late Teneral	80
	31–45	Early Mature	13
	45–59	Mature	7
	59–76	Early Senescent	0
	76–90	Senescent	1
15–19 August	0–15	Teneral	18
	15–31	Late Teneral	51
	31–45	Early Mature	24
	45–59	Mature	23
	59–76	Early Senescent	2
	76–90	Senescent	7

## Data Availability

Data generated or analyzed during this study are available from the corresponding author upon reasonable request.

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
