# Peer review of "Establishing Age-Based Color Changes for the American Burying Beetle, Nicrophorus americanus Olivier, with Implications for Conservation Efforts"

_insects, 2023, doi:10.3390/insects14110844_

Round 1

Reviewer 1 Report (Previous Reviewer 1)

Comments and Suggestions for Authors

The paper is much improved.  The study still describes the use of lab-reared beetles in several locations, although it appears that wild-caught beetles were added to the study.

321-350 still describes lab-reared insects without the inclusion of the wild-caught beetles.  This section contains material that does not speak to the objectives of the study.

399-410 can this be changed since wild-caught beetles have now been tested?

References are not in alphabetical order – if that is a journal requirement

Author Response

We thank the reviewer for re-reading our paper. We added data from field -collected beetles that were aged by comparing with our lab-based results.  The lab results created a tool that is simple to use and widely applicable for additional research on American burying beetles.  It is not possible to capture and hold beetles in the field while they age.

321-350. We used the laboratory captive beetles to photograph color changes for every other day from emergence to death. It is not possible to do the same in the field because captured beetles are of unknown age and because they are federally protected, they are released after capture.

399-410.  In order to generate the figure and show change over time, we have to analyze photographs from the laboratory beetles.

Reviewer 2 Report (Previous Reviewer 2)

Comments and Suggestions for Authors

Really the information about color changes in insects in relation to their age is very scarce. So reviewing work qualitatively filled up specific gap in this field. The authors with novel and rigorous methodology showed that the change in greyscale between the pronotum and elytra markings was significantly different, with more rapid darkening in the pronotum. Besides linear models were used to confirm authors conclusions about color changes in beetles body   when time series photography was assessed.

The authors carefully have revised the MS and I recommend the latter to be published.

Author Response

We thank the reviewer for reviewing our revisions and agree that this paper will make contributions to both American burying beetle conservation and to better understanding and documenting insect aging. 

Reviewer 3 Report (Previous Reviewer 3)

Comments and Suggestions for Authors

The recommendations of my previous review have not been fulfilled (redo the measurements with accurate spectrometer) then my opinion is not changed.

I understand that authors are working on an easy/fast way to take measurements in the field. But Accurate field spectrometers (plugged for example on an USB port of a laptop) exist and I continue to consider smartphones are not appropriate for accurate results.

Author Response

In response to reviews, we showed how our technique can be used in the field to better characterize age-structure of a federally protected insect species.  We added materials to our introduction to explain the purpose of creating a tool that is usable by conservationists without additional costs. We added information tot eh introduction (lines 212 to 218) highlighting spectrophotometry.  Most importantly, the literature (reference 13) found that spectrophotometric measures under field conditions are not practical or feasible ex situ.  The technology has advanced, but recent publications of which we are familiar, still kill insects being examined (field) or measure insects in highly controlled laboratory settings.

To address the reviewer concern, we added "As technology improves, the age-grading technique may be able to be improved using relatively inexpensive field-portable spectrophotometers [13]. However, any use of a spectrophotometer would need to be optimized to take a reading on a live insect that can quickly be released unharmed. Ultimately, we believe that a print or electronic color chart will allow researchers and conservationists to adopt better age-grading and easily estimate adult American beetle age in a nondestructive manner." to lines 905-911.

Recent papers using spectrophotometric measures in the laboratory exist (10.1093/beheco/arab133) and we can cite them, but, there is no similar study of which we are aware that allows field spectrophotometric measures that also allows portability (traps located at least 2.2 KM apart), positioning of live insects without harm for the measure, followed by rapid release.. 

We added a sentence to our discussion highlighting the potential for future improvements, "". 

Reviewer 4 Report (Previous Reviewer 4)

Comments and Suggestions for Authors

The authors have taken much effort to improve the manuscript, aided by further explanation of the methods.

There are just some minor issues to consider. 

Line 26 change to “and should allow”.

Line 169-170 change to “The red, green, blue (RGB) values were obtained for each beetle using open-source Fiji/ImageJ computer software developed by the U. S. National Institutes of Health.”

Comments on the Quality of English Language

fine 

Author Response

We thank the reviewer for assessing our revisions.  We have corrected the items noted below.

On line 19, we changed the text to "should allow conservation..."

Lines 287-289 was changed as suggested to "The red, green, blue (RGB) values were obtained for each beetle  using open-source Fiji/ImageJ computer software developed by the U. S. National Institutes of Health was used [15]."

We appreciate this much-better wording.

Round 2

Reviewer 1 Report (Previous Reviewer 1)

Comments and Suggestions for Authors

The paper still contains excessive material that does not speak directly to the objectives.  This content is subject to the discretion of the author.  The content of the paper that speaks to the objectives is acceptable.

This manuscript is a resubmission of an earlier submission. The following is a list of the peer review reports and author responses from that submission.

Round 1

Reviewer 1 Report

Comments and Suggestions for Authors

It doesn’t appear that your photography technique is well documented. The discussion references several limitations and optimizations that may help with digital photography methods but the methods don’t speak to that.

112 A little more discussion on photography methods would help. Camera field of view, distance from image, room illumination during photography, etc.  How was beetle pose chosen?  What are the 3 photography techniques referenced in line 154?  What was the color value of the “white background.

134 Were 25x25 pixel samples taken from full 50MP images, or were they downsampled?  What is dimension of 25x25 sample area?

139 How did you record a color histogram? Peak, average or some other value.  I presume the color values were on a 0-255 scale? Label y axis

Both the introduction and the discussion, while interesting, speak to much more than just the  objectives of the paper

312-322 seems beyond the scope of the objectives

The concept discussed in the paper is novel and the goal of providing assistance to breeders is noble.  The color scheme provided is unlikely to be discernable to the human eye without the aid of technology.  The authors’ candid discussion of the limitations of the technique and the need for technical enhancement may pave the way for continued refinement of this technique.

295-310 The fact that these were lab-reared beetles seems to be significantly limiting, even by the authors.  Why was it not applied to wild-caught beetles? 

Reviewer 2 Report

Comments and Suggestions for Authors

Really the information about color changes in insects in relation to their age is very scarce. So reviewing work qualitatively filled up specific gap in this field. The authors with novel and rigorous methodology showed that the change in greyscale between the pronotum and elytra markings was significantly different, with more rapid darkening in the pronotum. Besides linear models were used to confirm authors conclusions about color changes in beetles body   when time series photography was assessed/

The main note to the MS is the absence of similar studies in other insects in reference.  In addition, it would be desirable to highlight the applied aspect of such research to a greater extent.

Reviewer 3 Report

Comments and Suggestions for Authors

Reviewer 4 Report

Comments and Suggestions for Authors

This is an interesting study that examines the change in adult colour of the American burying beetle, Nicrophorus americanus, in efforts to improve age estimation of threatened populations. The study used a photography set up to capture unique colours in adults from teneral to senescent stages. It concluded that changes in grayscale near the pronotum were distinct and could be used in further studies for research into field estimations of beetle age.

The introduction is thorough in providing background to the beetle’s status, current programs and theory of colour change. Methods for maintaining beetle colonies and camera set up are given in enough detail.

There are some minor spelling and grammatical errors throughout this manuscript.

Specific comments

Line 14 “beyond” consider changing to “between and including” and then add “adult stages” after “(old)”.

Line 22. “at15-day” please inset a space between “at” and “15”

Line 23.  Consider in this sentence removing several “and” and replacing them with “,”  “The ability to more precisely estimate age will refine viable population estimates ,and improve laboratory breeding pro-grams ,and reintroduction attempts and monitoring”.

Line 48 “elytral” should this be “elytra”? or insert the name of appropriate part of the elytra.

Results

Line 190. In the table please insert a gap in the line between “pronotum and AIC” and “elytra and AIC” and make “elytra” bold, to clearly demarcate table sections.

Line 206. “17% faster” please elaborate what this value is being compared to i.e. is it the change in the different colours or the colour on a specific or several body parts of the insect?

Discussion

Line 225. The following sentence is results “On average (± SD), beetles lived 66 ± 24 days with five beetles living greater than 90 days”. The “66” has already been said before, just remove this sentence from here and please add the SE and “five beetles living greater than 90 days” back into lines 162-163.

Comments on the Quality of English Language

see suggestions under "Specific comments"